



# Surface circulation characterization along the middle-south coastal region of Vietnam from high-frequency radar and numerical modelling

Thanh Huyen Tran[1,2], Alexei Sentchev[1], Duy Thai To[3], Marine Herrmann[4], Sylvain Ouillon[2,4], Kim Cuong Nguyen[5]

[1] Laboratory of Oceanology and Geosciences (LOG), UMR 8187, Univ. Littoral Côte d'Opale, CNRS, Univ. Lille, IRD, Wimereux, France
[2] Land-Ocean-Atmosphere Regional Coupled System Study Center (LOTUS), University of Science and Technology of Hanoi (USTH), Hanoi, Vietnam
[3] Institute of Oceanography, Vietnam Academy of Science and Technology (VAST), Nha Trang, Vietnam
[4] Laboratory of Space Geophysical and Oceanographic Studies (LEGOS), UMR 5566, IRD, CNES, CNRS, UPS, Toulouse, France
[5] University of Science (HUS), Vietnam National University, Hanoi, Vietnam

*Correspondence to:* huyen-thanh.tran@univ-littoral.fr

**Abstract.** Coastal water dynamics along the Vietnam Middle-South Coastal region (VMSC), part of the South China Sea, is highly complex with large spatio-temporal variability whose drivers are not yet well understood. For the first time, high-resolution surface current data from high-frequency radar (HFR) measurements were obtained in this region during the early (transition) phase of the Asian summer monsoon. The data were used to compare with simulation results from a circulation model, SYMPHONIE, and ultimately to optimise the wind forcing in the model. Both modelling and HFR were able to show the spatial and temporal evolution of the surface circulation, but some discrepancies were found between model and HFR data on some days, coinciding with the evolution of the wind. Two methods were used to optimise the wind forcing, namely the Ensemble Perturbation Smoother (EnPS) and the wind correction method using wind-driven surface currents (EkW). Both methods achieved a significant reduction (~36-40 %) in the error of the surface current velocity fields compared to the measured data. Optimised winds obtained from the two methods were compared with satellite wind data for validation. The results show that both optimisation methods performed better in the far field, where topography no longer affects the coastal surface circulation. The optimisation results revealed that the surface circulation is not only driven by winds but also by other factors such as intrinsic ocean variability which is not entirely controlled by boundary conditions. This indicates the potential usefulness of large velocity datasets and other data fusion methods to effectively improve modelling results.

## 1 Introduction

The coastal waters of Vietnam are parts of the South China Sea (SCS) ("East Sea" in Vietnamese term), one of the world's largest marginal seas, where various meteorological and oceanographic processes contribute to its circulation variability at



different scales. The region is under the influences of two monsoonal wind regimes: the southwest monsoon usually starts from June while the northeast wind usually begins from late November and fully developed in January (Potemra and Qu, 2009; Wang et al., 2021). The large scale circulation in the SCS is mainly driven by winds (Potemra and Qu, 2009), resulting

in seasonal patterns and phenomena which have been documented in many studies: SCS western boundary current (Fang et al., 2012), a dipole structure characterized by cyclonic and anti-cyclonic gyres offshore Vietnam (Chen et al., 2012; To Duy et al., 2022; Wang et al., 2006), and a South Vietnam upwelling (SVU) which usually occurs in summer (Chen et al., 2012; Da et al., 2019; To Duy et al., 2022; Herrmann et al., 2023, 2024). In particular, along the middle-south coasts of Vietnam (hereafter called VMSC), shown in Fig. 1b, all the three features of circulation mentioned above are present. Besides, the

coastal dynamics along the VMSC is strongly influenced by ocean intrinsic variability (OIV) (Li et al., 2014; To Duy et al., 2022; Da et al., 2019; Herrmann et al., 2023). These findings have emphasized that the dynamics of the VMSC is highly complex with large space and time variability governed by different factors, which are still not well understood.

The limitation of existing studies in the SCS, in particular along the VMSC, is that their research mostly relied on coarse-resolution modelling, intermittent satellite data and discrete measurements. Furthermore, it is worth noting that, the majority

of recent studies in the region mainly focused on the ocean dynamics and circulation variability during the matured monsoon periods, meanwhile, ocean dynamics during the transition periods is still lacking of knowledge. During the transition period, the wind changes direction in complex ways thus affecting the temporal and spatial variability of the surface currents. These, among others, can generate various dynamic structures i.e. submeso-scale eddies, divergence and convergence zones. Modelling approach, to some extents, can help to acquire a general understanding of the ocean circulation with sufficient and

reliable forcing inputs. However, for Vietnam coastal regions, atmospheric forcing data, especially winds, are very limited and often come from global data sources with low spatial resolution, thus, some dynamic features at local/regional scales can be poorly presented or even neglected. Nguyen-Duy et al. (2023) have conducted sensitivity tests of Red River plume dynamics to wind fields in the Gulf of Tonkin. The river plume was found to be sensitive to wind variations which generate a clear reversal of the circulation pattern and affect the plume thickness variability. Meanwhile, as was demonstrated in our

preliminary work, during the transition period of the summer monsoon from April to May, the VMSC coastal dynamics experienced a high variability both in terms of temporal and spatial scales associated with large variability in wind fields (Tran et al., 2022). Thus, in order to deliver a comprehensive understanding of the coastal dynamics during the monsoonal transition period, high-resolution datasets and better, reliable model simulation results are necessary.

The questions now are, how high-resolution measurements can be used and how numerical models can be improved for better

simulating fine-scale processes in coastal regions, in particular, along the VMSC? In the past years, drifters and satellite-derived data have been used in many studies as the sources of observation data. Centurioni et al. (2009) used Surface Velocity Program drifters with reanalysis wind and satellite altimetry data to construct 2D circulation maps in the SCS at 1° resolution. Similar approach has been employed for estimating mean geostrophic currents in the SCS with Argo profiling floats and altimeter data (Yang et al., 2019; Zhou et al., 2010). Nonetheless, these products only provide estimated or derived sea surface

currents (SSC). Up-to-now, no direct measurements with high-spatio-temporal resolution are available for the VMSC. High-



frequency radar (HFR), a land-based remote sensing technique, has been widely used  in oceanographic studies since the system has the ability to monitor SSC directly with high temporal and spatial coverage (Mantovani et al., 2020). The capability of HFRs in monitoring surface currents in the tropic coastal regions have been proven in a number of existing studies within a wide range of oceanographic applications, e.g. coastal circulation characterization (Cosoli et al., 2020), assessing the
turbulence dispersion of passive tracers and particulate materials (Tran et al., 2022), and coastal circulation response to climatic variabilities (Gu and Mao, 2024). On the Vietnam coasts, two different radar systems have been installed: three CODAR Seasondes at ~6km resolution in the Gulf of Tonkin, north of the SCS (Rogowski et al., 2019); and two WavE RAdars (WERA) at 1km resolution in the VMSC. Our preliminary results showed that HFRs could demonstrate well the fine-scale structure of coastal circulations during the transition monsoon period (Tran et al., 2022). In terms of modelling, in the most recent studies
of Herrmann et al., (2023, 2024); To Duy et al., (2022), high-resolution circulation model SYMPHONIE was used to investigate the interannual variabilities of SVU in the SCS including the VMSC. However, no current velocity measurements have been used for model validation or optimization.

There are several approaches in model forcing optimization, for example, correcting the model boundary conditions (tides and winds). In this method, sets of perturbed wind and tidal forcing ensembles were generated with inverse Fast Fourier Transform
(FFT) method. Subsequently, tidal and wind forcing were optimized via data assimilation techniques (Barth et al., 2009, 2011; Marmain et al., 2014). In the Gulf of Tonkin, European Center for Medium-Range Weather Forecasts (ECMWF) wind forcing was perturbed by using Empirical Orthogonal Functions (EOF) analysis for model cluster analysis (Nguyen-Duy, 2022). Differing from these two methods, our research introduces a physical approach which quantifies the surface current component induced by winds using HFR data, without ensemble simulation. The method allows to optimize the wind forcing input of the
3D-ocean circulation model SYMPHONIE with the expectation to improve the capabilities of the model in simulating fine-scale coastal current structures of the VMSC during the onset of the summer monsoon.

## 2 Methodology

### 2.1 HFR measurements in the VMSC region

HFR system used in the study is a pair of WERA ocean radars with the central operating frequency of 16.15MHz.  Radial
surface current velocities were measured by two radar stations (denoted as V1, V2 in Fig. 1c) performed during the measurement campaign in Phu Yen, VMSC from 14th of April to 16th of May, 2019. The measurement domain spanned from [12° 53' N, 109° 16' E] to [13° 22' N, 109° 47' E] with a spatial resolution of 1km. The raw radial velocities were treated with RMSE-threshold filtering in order to be free of spikes. Surface current velocity vectors were then reconstructed from the noise-removed radial velocities with the 2DVar/EOF method (Yaremchuk and Sentchev, 2009, 2011). The interpolation and gap
filling procedure has been explained in our previous work (Tran et al., 2022).



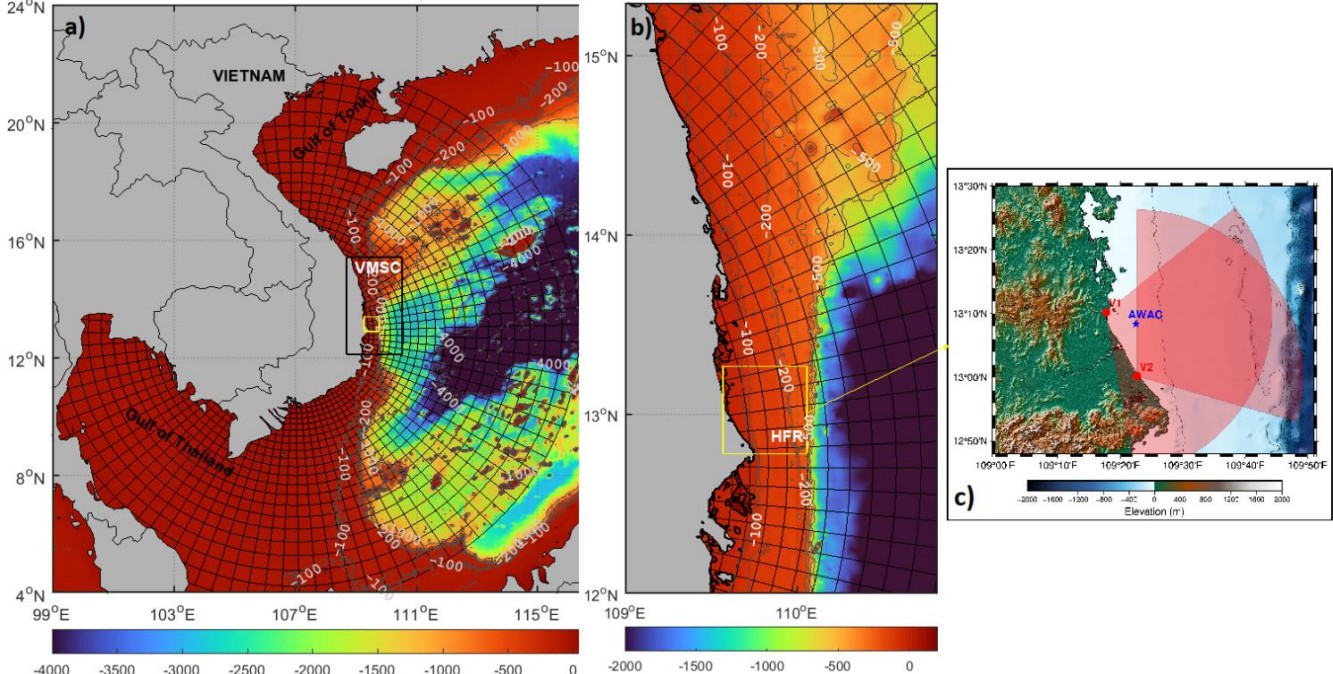

**Figure 1 a) Bathymetry map of the EASTSEA domain and b) the VMSC domain (right) (bathymetry data source: GEBCO_2022). VMSC domain is bounded with black box, HFR domain bounded with yellow box. The computation orthogonal curvilinear grid is illustrated by black solid curves (the grid node density was reduced for better demonstrative purpose); c) Spatial coverage by WERA radars installed in April-May 2019 at the Phu Yen coasts, VMSC. Red dots show the two HFR stations V1, V2. Blue star shows location of AWAC. Bathymetry contours are shown by black solid lines.**

Surface currents from HFR have been validated with 7-day surface current velocity measurements from AWAC (Acoustic Wave And Current Profiler). Based on three metrics of evaluation: correlation (R), root-mean-squared error (RMSE) and mean absolute error (MAE) of surface currents, SSC time series obtained from HFR and AWAC were found to be in a good agreement (R=0.58, Root-mean-squared error RMSE=0.10, Mean absolute error MAE=0.07 for u- and R=0.72, RMSE=0.13, MAE=0.11 for v-components) (Fig. 2).



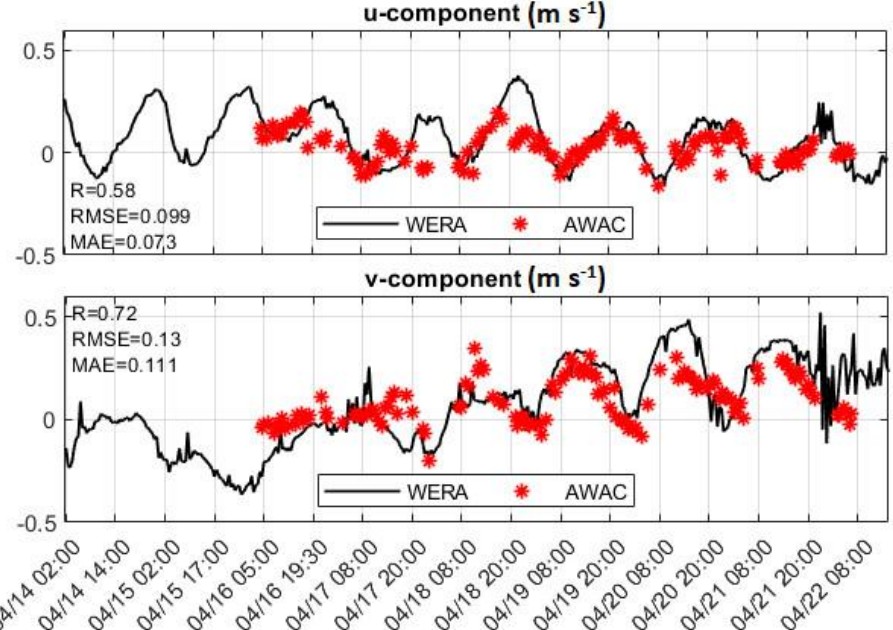

**Figure 2:Comparison of SSC time series from HFR (black solid line) and AWAC (red asterisk points)**

## 2.2 3D-circulation model SYMPHONIE

The 3D ocean circulation model SYMPHONIE is a Boussinesq hydrostatic ocean circulation model developed by the SIROCCO group (CNRS & Toulouse University, France). Based on an Arakawa curvilinear C-grid, momentum and tracer fluxes are computed using an energy conserving finite difference method (Marsaleix et al., 2008). The model adopts the time stepping method, using Leap Frog scheme combined Laplacian filter; K-epsilon turbulence closure scheme, and pressure gradient Jacobian scheme to describe the physics within the model. Description of various boundary conditions including lateral open boundaries, rivers, surface conditions can be found in (Estournel et al., 2009).

The model configuration used in this research was adopted from the regional model configuration described and used in Duy et al., (2022); Herrmann et al., (2023, 2024); and Trinh et al., (2024) . The grid resolution is ~1 km nearshore and coarsens gradually to ~4.5 km in the open seas. The atmospheric forcing is introduced via the bulk formulae of Large and Yeager (2004) using the 3-hourly output of the ECMWF 1/8° atmospheric analysis. Initial ocean conditions and lateral ocean boundary conditions are prescribed from the daily outputs of the global ocean 1/12° analysis PSY4QV3R1 from the Copernicus Marine and Environment Monitoring Service (CMEMS). Tidal data were taken from the 2014 release of the finite-element solution (FES2014) global tidal model. Nine tidal constituents have been taken into account. River forcing was enabled with the information of river systems along the coasts and was described in To Duy et al. (2022).

The model domains are shown in Fig. 1. The first simulation started from January to June, 2019 on the entire SCS domain (Fig. 1a) (EASTSEA run) in order to obtain open boundary conditions (OBC) for the VMSC domain (Fig. 1b). The second



simulation (VMSC run) has been conducted in the VMSC domain from March 1 to May 5, 2019 for generating initial conditions before the optimization procedure was executed. The VMSC domain has the same grid resolution as the EASTSEA

domain. Details of the experiment setup are described in Sect. 2.4.

### 2.3 Methods of wind forcing optimization

#### 2.3.1 Ensemble perturbation smoother (EnPS)

The wind optimization procedure adopted from (Barth et al., 2011; Marmain et al., 2014) consists of two steps: (1) ensemble perturbation and (2) data assimilation. The period of analysis was set from 5 to 15 of May, 2019.

A total of 30 ensembles were generated during a period of two months (April-May, 2019) by using wind fields with perturbation data.

The perturbations ($\boldsymbol{w}_p(x,y,t)$) were computed from the real part of a quantity which are the sum of the multiplication of a random complex normal-distribution time-series ($z_k(t)$) and decomposed wind fields ($\boldsymbol{a}_k(x,y)$) obtained from Fast Fourier Transform (FFT):

$$\boldsymbol{w}_p(x,y,t) = \alpha \, \mathrm{Re}(\textstyle\sum_{k=1}^{N} \boldsymbol{a}_k(x,y) \, z_k(t)) \tag{1}$$

N is the total time-series length of wind data during the two months. α is the scaling factor between the final expected wind error and temporal error. With this method of perturbation, the spatial and temporal distributions of wind perturbation errors were assumed to be similar to that of ECMWF winds in order to avoid errors induced by nonlinear variabilities.

EnPS is a non-sequential method where all observations are used to generate optimal state (an analysis) for a specific DA time

window. The analysis of wind $\boldsymbol{w}^a$ is yielded from the equation:

$$\boldsymbol{w}^a = \boldsymbol{w}^b + \boldsymbol{w}^{incr}, \tag{2}$$

where $\boldsymbol{w}^b$ is the background wind, $\boldsymbol{w}^{incr}$ is the increment added to the background field. This increment can be achieved from the relationship of a number of quantities: the covariance matrix of wind ensembles (**S**), the covariance matrix of so-called "observation operators" that link model surface currents with wind forcing (**E**), the observation error matrix **R**, and the

difference between observations $y^o$ and the model operators $h(\boldsymbol{w}^b)$ . Hence, the Eq. (2) can be rewritten as:

$$\boldsymbol{w}^a = \boldsymbol{w}^b + \mathbf{S}\mathbf{E}^T(\mathbf{E}\mathbf{E}^T + \mathbf{R})^{-1}(y^o - h(\boldsymbol{w}^b)), \tag{3}$$

in which, $\mathbf{S}\mathbf{E}^T$ represents the covariance matrix between wind forcing and model (surface currents), while the covariance matrix of model surface currents is represented by $\mathbf{E}\mathbf{E}^T$:

$$\mathbf{S}\mathbf{E}^T = cov(\boldsymbol{w}^b, h(\boldsymbol{w}^b)) \tag{4}$$

$$\mathbf{E}\mathbf{E}^T = cov(h(\boldsymbol{w}^b), h(\boldsymbol{w}^b)) \tag{5}$$

The detailed explanation of the DA scheme and the components of the formula can be found in Barth et al. (2009, 2011).



In this research, we conducted DA for a 10-day long analysis period (May 5-15, 2019) with a time window of 24-hours. For each DA run, 8 instances of wind data (3h interval) corresponding to 24 instances of surface current data (1h interval) were used.

### 2.3.2 Wind correction from wind-driven surface currents (EkW)

We assume that the uncertainty of model surface current simulation comes only from the inaccuracy of wind forcing without consideration of other sources of uncertainties (i.e. open boundary conditions, initial conditions and other forcing fields). In other words, the wind-driven component of surface currents was not properly reproduced in the model due to coarse-resolution wind data. This uncertainty can be compensated by a correction of model surface currents ($\boldsymbol{u}_{cor}$) represented by the difference between observations, i.e., surface currents obtained from HFR ($\boldsymbol{u}_R$) and model ($\boldsymbol{u}_M$):

$$\boldsymbol{u}_{cor} = \boldsymbol{u}_R - \boldsymbol{u}_M \tag{6}$$

For each value of wind data, surface currents at neighbouring points located within a certain distance $R$ to the wind data point were considered. In order to validate the method later with the same measurement data from HFR, only 60 % of the data points in HFR current field were randomly selected for the wind forcing correction. The remaining 40 % of data were used for evaluation of the optimization method skill. The correction of wind speed $\boldsymbol{w}_{cor}$ was estimated from the surface current velocities using the empirical formula of (Weber, 1983):

$$|\langle \boldsymbol{u}_{cor}(x_i, y_i, t) \rangle| \approx 27 \sqrt{C_d \frac{\rho_a}{\rho_0}} |\boldsymbol{w}_{cor}(x, y, t)| = \beta |\boldsymbol{w}_{cor}| , \tag{7}$$

where $C_d = 1.6 \times 10^{-3}$ is a constant drag coefficient, $\rho_a$ and $\rho_0$ are the air and water densities respectively, and $\langle \boldsymbol{u}_{cor}(x_i, y_i, t) \rangle$ indicates the spatial average of velocities over a circular area with a radius R = 13 km (~0.125 degree). As the temporal and spatial resolution of wind and surface current velocities are different, we applied a least-squares method to calculate 3h analysis wind from a series of 1h surface current velocities. The cost function in least-squares method is henceforth described in the following equation:

$$J(\boldsymbol{w}_{cor}) = (\langle H(\boldsymbol{u}_R - \boldsymbol{u}_M) \rangle - \beta \boldsymbol{w}_{cor})^2 \to min \tag{8}$$

Here, $H$ is an operator projecting the current velocities in locations $(x_i, y_i)$ onto the wind vector location. The corrections of wind were finally added to the background wind field to obtain the analysis wind as in Eq. (2) of the EnPS method.

### 2.4 Experiment design

Since the focus of the paper is the VMSC region, we implemented the optimization steps only for the VMSC domain in order to reduce the cost of computation. Before conducting optimization procedure, OBC and initial states of the model have to be generated. The first simulation was made for the entire EASTSEA domain to obtain OBC for the VMSC domain. Subsequently,



initial states were created from the VMSC run. Three simulations, VMSC_ref, VMSC_EkW and VMSC_EnPS, used the same initial states provided by the VMSC run on May 5, 2019, but with three different wind forcing conditions (Table 1).

**Table 1 Model run description**

| Model run | Period | Description |
|-----------|--------|-------------|
| **EASTSEA** | Jan 1-June 1, 2019 | Generate OBC for VMSC domain |
| **VMSC** | Mar 1-May 5, 2019 | Spin-up run for OBC to be propagated into the VMSC domain |
| **VMSC_ref** | May 5-May 14, 2019 | Reference run (free run) |
| **VMSC_EkW** | May 5-May 14, 2019 | Simulation with EkW method |
| **VMSC_EnPS** | May 5-May 14, 2019 | Simulation with EnPS method |


Skills of the two optimization methods were evaluated by using RMSE and MAE of surface currents to compare the remaining 40% of the HFR measurement data with modelling results at corresponding point locations. For validating the performance of the two optimisation methods, wind data from Advanced Scatterometer Data Products (ASCAT) at 0.25-degree resolution were used to compare with EnPS and EkW wind data.

**3 The VMSC hydrodynamics characterization during the Monsoon transition period (April-May)**

**3.1 Spectral analysis**

Spatially-averaged power spectral density (PSD) has been computed for the surface current velocity time-series obtained from HFR and VMSC_ref as well as for the wind velocity time-series from ECMWF during the measurement period (Fig. 3). In all cases, the power spectra have a similar shape for both cross-shore and along-shore dimensions. There are two prominent peaks
in PSD of wind associated with inertial and diurnal periods. The diurnal cycle of winds is governed by the land-sea thermal exchange between day and night due to high difference in surface heating. Regarding PSD of current velocity, a frequency of ~0.04 hour$^{-1}$ (24 h period) associated with diurnal tidal motions are found in both VMSC_ref and HFR, however, the level of energy differs considerably between the two time series. Lower-frequency band processes were also detected during this period of analysis including inertial motion caused by Coriolis force. Regarding the inertial period, a significant peak was found for
VMSC_ref's time-series associated with wind time-series, but not for HFR measurements. The energy levels of VMSC_ref and HFR power spectra are comparable in the low frequency band until approximately 0.025 hour$^{-1}$ (~40 h period) then the energy in VMSC_ref's power spectra declines drastically in high-frequency band compared to that in HFR measurement. Meanwhile, on the left side of the spectra, there are some slight shifts at longer periods (more than 1.5 days) between along-shore and cross-shore spectra. At synoptic scale (several days), wind and HFR spectra have a peak at a frequency band of





approx. 0.0095 hour$^{-1}$ (4-5 days), however the process in surface current show to be more resistant (wider peak) in time compared to that of wind (sharper peak). This pattern is not clearly observed in the spectrum of VMSC_ref's current velocity.

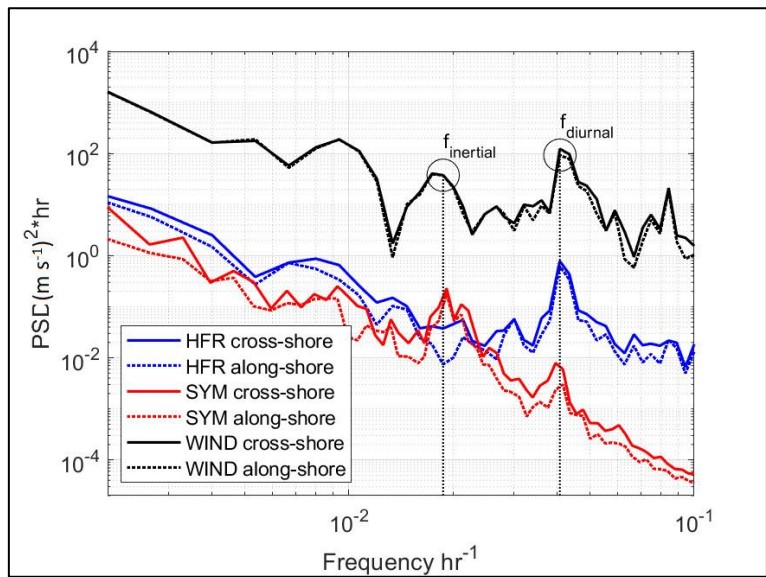

**Figure 3: Power spectral density (PSD) of ECMWF wind velocity (black), SSC velocities from HFR (blue) and VMSC_ref (red)**


Differing from the PSD spectra of VMSC_ref's SSC that shows a clear energy cascade from larger to smaller scales, the steep slope at high-frequency bands is not observed in those of wind and SSC obtained from HFR measurements. In case of wind, the coarsely spatio-temporal resolution of wind data can cause this behavior in PSD spectrum. Meanwhile, since SSC fields from HFR and VMSC_ref are comparable in time and space (1 h and ~1 km resolution), the high level of energy at the high-

frequency bands could come from the noise level in the HFR measurement data and the accuracy of currents from HFR measurements. The high noise level in HFR data was found where the 50 Hz-noise harmonic bands induced by power supply were located. Figure A1 in the Appendix will explain this effect.

### 3.2 Tidal analyses

The tidal components of surface currents were estimated for the measurement period using "Unified Tidal Analysis and

Prediction" Matlab functions (Utide) developed by the University of Rhode Island (Codiga, 2011). The tidal regime in the region is predominately diurnal with three main tidal constituents: K1, P1 and O1. There are some differences in tidal ellipse spatial distributions between VMSC_ref and HFR. The HFR tidal ellipse maps show that clockwise rotation ellipses are observed near the coast, whereas counter-clockwise ellipses are located offshore and in the southeastern part of the domain. The shape of ellipses is more circular nearshore and more elongated seaward. The velocity magnitude is also larger offshore

than nearshore (Fig. 4).



Meanwhile, this elongation of ellipses offshore is not visible in VMSC_ref. The magnitude of tidal currents from VMSC_ref is overall smaller than that from HFR. The better agreements between VMSC_ref and HFR are found in the middle field of the HFR domain (13-13.2º N, 109.4-109.6º E). Figure A1 in Appendix shows higher uncertainties of both u and v components of SSC from HFR measurements are found in the offshore range starting from a distance of ~40 km. This pattern coincides

with the discrepancies in tidal ellipse distribution maps between HFR and VMSC_ref.

**Figure 4: Tidal ellipse distribution for P1 constituent (left) and K1 constituent (right) from HFR (top row) and from VMSC_ref (bottom row)**





**3.3 SSC variability**

Before analysing the results during the analysis period (April-May, 2019), we investigate the multiannual variability of SSC field during the same months in the past ten years. SSC during April-May in the years 2009-2018 were simulated by SYMPHONIE for the EASTSEA domain. The 10-year average of the SSC field over these two months is shown in Fig.5a, b. It clearly shows a seaward current occurring within the latitude of 10-14ºN. This dynamic feature results in an associated upwelling which starts to form offshore of the VMSC during summer.

For the current period of analysis (April-May, 2019), the Fig. 6 shows a remarkable consistency between HFR and EASTSEA run. An intense flow at the southeastern part of the HFR domain has been detected (Fig. 6a, d). This was influenced by the effects of the large-scale South China Sea West Boundary Current (SCSWBC) represented by a strong northeast current. As the monsoon wind changes, the SSC field has a bimodal behavior, where the trace of the SCSWBC in winter (strong current with south-southwest direction) and some features of summer traits (strong jet with eastward-northeastward direction) were both observed in HFR data and well-presented in EASTSEA run, respectively end of April and middle of May (Fig. 6a-b). The weakening of winter SCSWBC together with the strengthening of Southwest monsoon wind caused a seaward flow which were captured in both EASTSEA run and HFR (Fig. 6c). During calm wind conditions, the southward current prevailed over the HFR domain (Fig. 6b). Meanwhile, the currents change the direction to northward and northeastward-eastward direction as southwest wind became stronger (Fig. 6a, c).





**Figure 5: a) Surface circulation in the western SCS in April-May, from the EASTSEA run. The current velocity fields have been averaged for the 10-year period, 2009-2018. b) A zoom-in to the VMSC domain. The colour scale depicts the velocity of surface currents in ms⁻¹. A large upwelling and current separation already occur at latitudes 10-14ºN during this time of the year (Tran et al., 2022)**



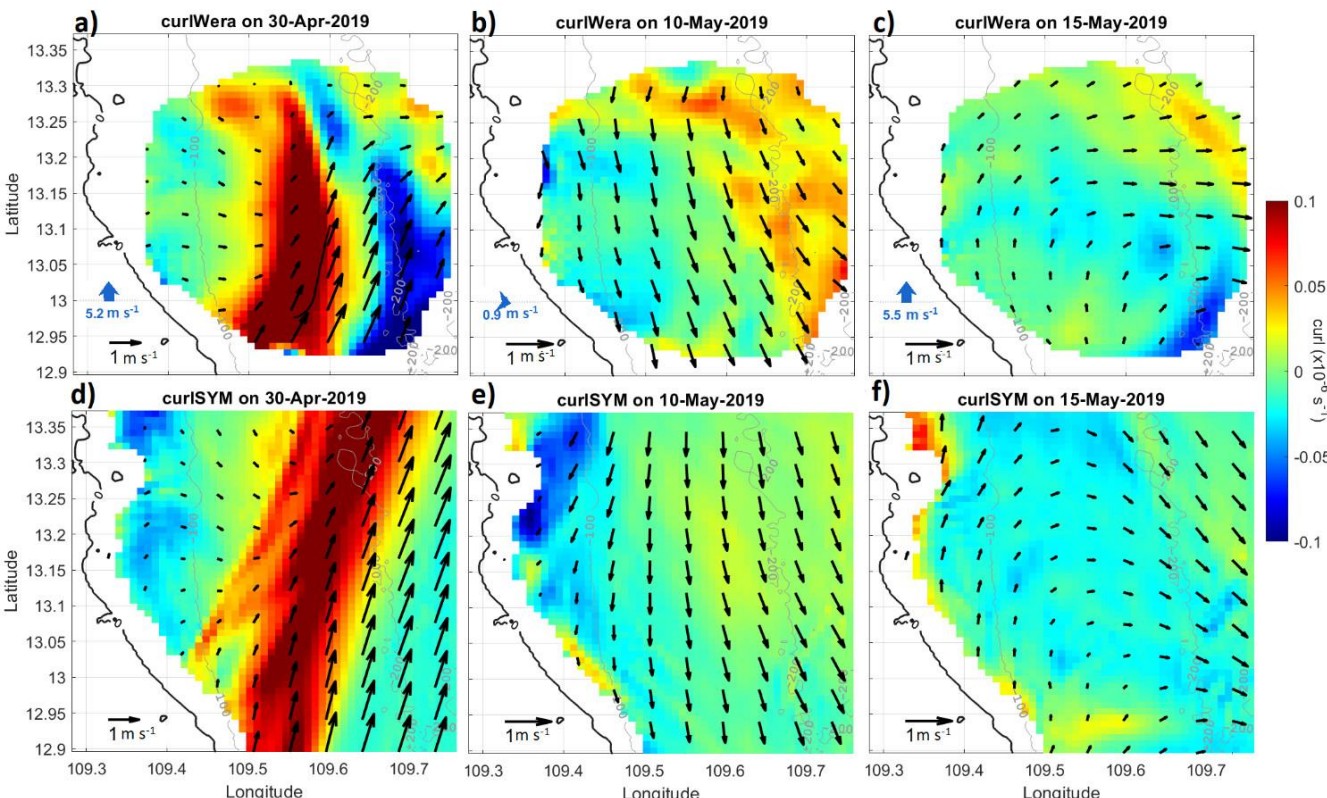

**Figure 6: SSC vectors obtained from HFR (a-c) and SYMPHONIE (EASTSEA run) (d-e) superimposing curl of SSC field. Blue arrows in the southwest corner denoted averaged wind directions and magnitudes over the HFR domain. Isobaths are given in gray solid lines, coastlines are represented in black solid lines.**

Two different evolutions were observed in both time-series of HFR and the two model runs (EASTSEA and VMSC_ref) (Fig. 7). From the mid-April until the first week of May, strong positive meridional currents were observed in the mid and far fields of the domain, whereas, at the end of the measurement period, the v component of SSC was inverted to negative and smaller values. This behavior coincides with the evolution of wind time-series, where wind started to change its direction from south-southeast to south-southwest. The magnitudes of SSC v-component are found to be strongly correlated in time with wind. However, a large discrepancy in time-series of v-component of SSC were spotted between EASTSEA, VMSC_ref and HFR at the first 20 days associated with south-southeast wind. During south-southwest wind conditions, the EASTSEA results show better agreement with HFR. In addition, the zonal component of SSC was generally underestimated and the meridional component, on contrary, was overestimated in VMSC_ref and EASTSEA compared to those derived from HFR measurements (Fig. 7a-c). This could come from the uncertainty of the wind forcing input used in the EASTSEA and VMSC_ref, where the impact of local effects is not well described in the global atmospheric models.



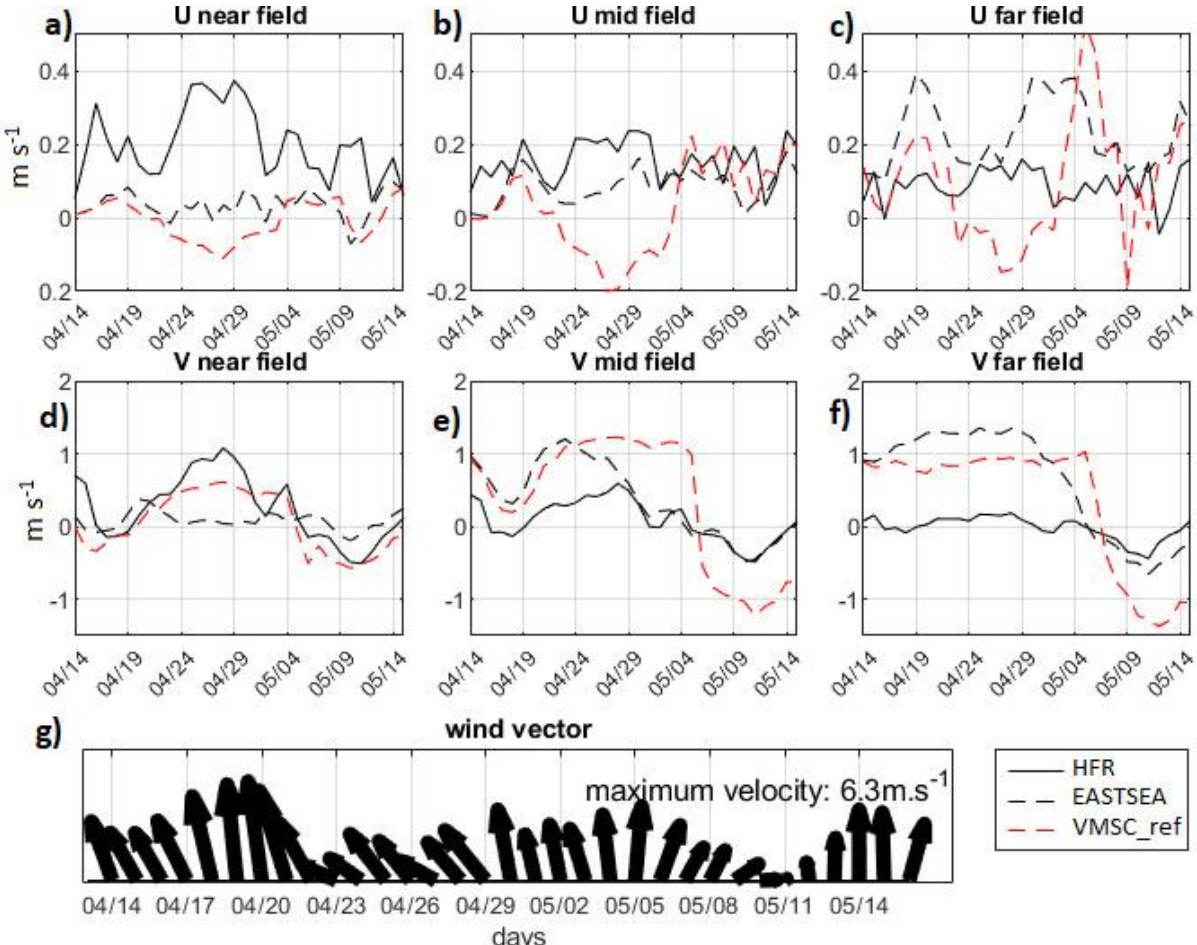

**Figure 7: Time-series of u and v components of SSC from HFR, EASTSEA and VMSC_ref (a-f); vector time-series of ECMWF wind**
**(g)**

The time series of u and, in particular, v components of SSC show significant differences between EASTSEA and VMSC_ref despite a similar trend was observed. The two runs started deviating after the third week of April (Fig. 7a-f), associated with a dramatic change in wind direction from strong southern wind to weak eastern wind (Fig. 7g). After the first ten days of May,
the u component of EASTSEA and VMSC_ref show a better agreement, whereas, the VMSC_ref still underestimated the v component of SSC compared to EASTSEA. The cause for this deviation will be discussed in Sect. 5.

**4 Evaluation of modelled surface currents resulting from wind forcing optimization**

The SSC velocities from VMSC_ref, VMSC_EkW and VMSC_EnPS are compared with SSC velocities obtained from HFR measurements. The vector difference map in Fig. 8 reveals that the EkW method seems to deliver a better result in vector error





reduction in the middle of the domain, as the SSC vector differences in VMSC_EkW (red arrows) are substantially smaller than those in VMSC_EnPS (blue arrows).

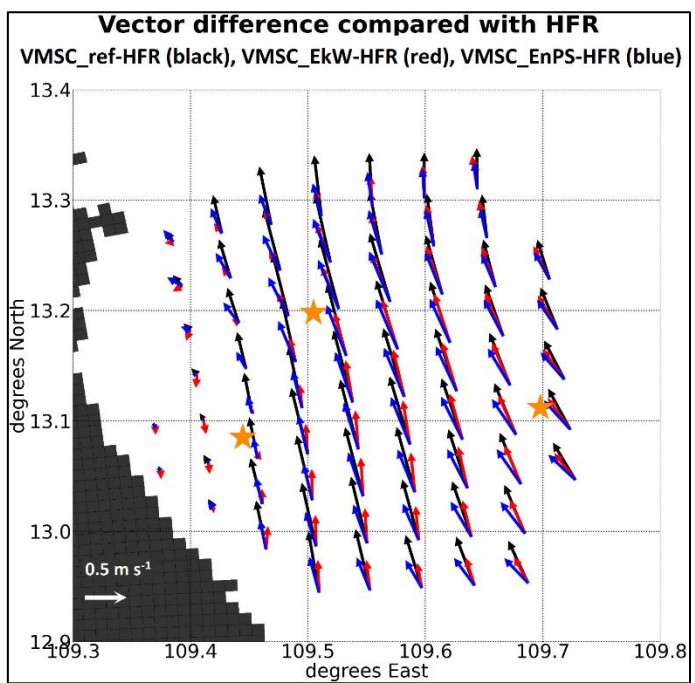

**Figure 8: Vector differences of SSC (temporally-averaged over the analysis period) between three simulations compared to HFR**
**measurements: VMSC_ref-HFR (black arrow), VMSC_EkW-HFR (red arrow), VMSC_EnPS-HFR (blue arrow). Locations of three selected points are denoted by orange stars. Vectors were plotted with a three vector point spacing.**

In order to evaluate the temporal evolution of velocity fields resulting from different runs (VMSC_ref, VMSC_EnPS and VMSC_EkW) more comprehensively, three points located at near field, mid field and far field of the HFR domain have been 300 selected for comparison (Fig. 9). Besides, three subdomains representing near-, mid- and far fields, where radar measurements were available, were also defined. Near field is limited within 109.3-109.45 degrees East, mid field within 109.45-109.6 degrees East and far field within 109.6-109.75 degrees East.

At near field point, the SSC vectors started rotating from the 2nd day of the period (6th of May, 2019) to the SE direction and had a similar evolution to that of HFR. Meanwhile, at mid and far field points, the SSC time-series during the first 2-3 days of 305 the analysis period show no significant response to the change of wind fields. This time lag corresponds to the slow adaptation of surface currents to wind forcing.





**Figure 9: Vector time series of wind (reference, EnPS, EkW) and SSC (VMSC_ref, VMSC_EnPS, VMSC_EkW and HFR) of three points located at near-, mid- and far-field of the HFR domain.**





ASCAT wind time-series at near-, mid- and far fields were extracted for comparison with EnPS and EkW winds (Fig. 10). The EnPS wind speed time-series has a better agreement with ASCAT wind compared to that in EkW (Fig. 10a-c). Wind speed at mid and far fields was improved with EnPS method during May 11-13 (Fig. 10b-c) when the wind blew in northward to northeastward direction (Fig. 7g). Figures 10a-c also reveal that the windspeed at the middle of the analysis period (especially during calm wind conditions) is overestimated in EkW method, whereas, EnPS wind speed curves evolved in the similar

manner compared to ASCAT.  In terms of v component of wind, EkW seems to achieve better fit to ASCAT at far field (Fig. 10f). The discrepancy between both methods and ASCAT started to increase from May 13 (the grey box on the right of Fig. 10a-f), when the wind speed started to increase considerably (Fig. 7g). The two methods seem to perform well at the middle of the analysis period (May 8-13).

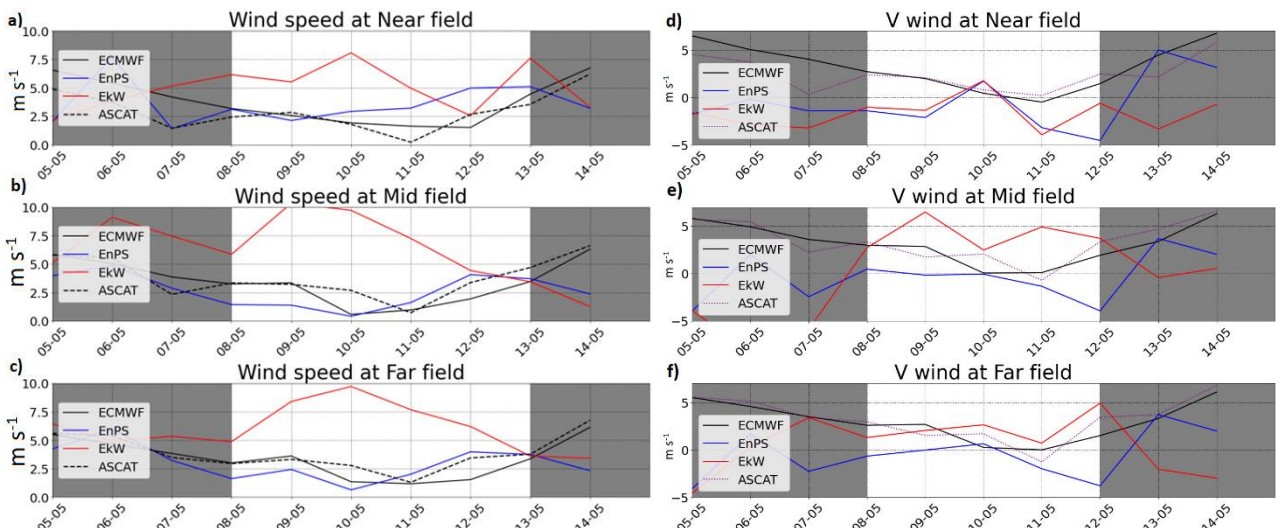


**Figure 10: Columns: Wind speed time-series from different wind data (left column); Time series of V-component of wind from wind data (right column); Rows: three sub-domains: near field, mid field and far field**

Spatially-averaged RMSEs of surface currents between VMSC_ref, VMSC_EkW and VMSC_EnPS against HFR were

calculated for the three subdomains. As a result, the RMSE time-series of u component of the SSC show no clear improvements from both optimization methods (Fig. 11). Meanwhile, for v component, the RMSE in VMSC_EkW was reduced significantly (on average, 25 % and 46 % for mid and far fields, respectively) during the analysis period. At the first seven days, the RMSE at near field in VMSC_EkW (dash-line) was found higher than that in the VMSC_ref, however, the RMSE dropped below the reference RMSE line (solid) from the latter days onward. For evaluating the optimization performance between EkW and

EnPS methods, spatially-averaged RMSE and MAE for the three subdomains are used as the evaluation metrics and summarized in Table 2. In most cases, EkW and EnPS method errors are reduced compared to the reference run. Both EkW and EnPS are able to mitigate the errors in v component of SSC significantly, however, only EkW method can achieve the





error reduction in u component of SSC. The subdomain which has the most remarkable improvement after optimization procedure is the mid field (36-46 % of error reduction).




**Figure 11: RMSE of u and v components of SSC during the analysis period from different simulations, spatially averaged over three subdomains**



**Table 2: Time- and space-averaged RMSE and MAE values (in m s⁻¹). Statistics were based on 40 % of measurement data which were not used for optimization procedure**

|  | **U** ($m\ s^{-1}$) | | | **V** ($m\ s^{-1}$) | | |
|---|---|---|---|---|---|---|
|  | *VMSC_Ref* | *VMSC_EnPS* | *VMSC_EkW* | *VMSC_Ref* | *VMSC_EnPS* | *VMSC_EkW* |
| **MAE** | | | | | | |
| *Near field* | 0.09 | 0.09 | **0.05** | 0.19 | **0.12** | **0.10** |
| *Mid field* | 0.17 | **0.16** | **0.08** | 0.74 | **0.46** | **0.38** |
| *Far field* | 0.20 | 0.23 | **0.17** | 0.59 | **0.40** | **0.44** |
| *Overall* | 0.17 | 0.17 | **0.10** | 0.58 | **0.37** | **0.35** |
| **RMSE** | | | | | | |
| *Near field* | 0.09 | 0.11 | **0.06** | 0.25 | **0.16** | **0.12** |
| *Mid field* | 0.18 | 0.18 | **0.10** | 0.74 | **0.47** | **0.40** |
| *Far field* | 0.22 | 0.24 | **0.18** | 0.60 | **0.41** | **0.45** |
| *Overall* | 0.18 | 0.19 | **0.13** | 0.63 | **0.40** | **0.38** |

## 5 Discussion

For the first time, high-resolution surface current velocity data were obtained from HFR measurements along the VMSC. The radar coverage represents approx. 4.5 % area of the VMSC domain configured in the model, where the interaction of sea and land as well as other small-scale coastal processes are in place. These processes are typically not well represented in model simulations due to the lack of high-resolution model forcing data and high-resolution bathymetry. The measurements revealed significant differences between model and data, despite temporal evolutions of all time-series (HFR, EASTSEA and VMSC) showed similar trends. Interestingly, surface current velocity fields obtained from EASTSEA and VMSC runs show large variations over time and also a significant difference during some periods of time. This discrepancy could be due to the deviation rising from the forcing propagation at the open boundaries to the smaller-size model domain of computation and the nonlinear chaos within the ocean body without altering atmospheric forcing conditions. In a previous study, in the northwestern Mediterranean Sea, Marmain et al. (2014) has concluded that unexpected features of ocean dynamics could come from a small inconsistency at exterior areas, as considering a domain of interest where the ocean dynamics is strongly influenced by large open boundaries. To support the assumption, we investigated the SSC field using April-May outputs of the SYMPHONIE ensemble simulation performed and analyzed by Herrmann et al. (2023) over the EASTSEA domain for the years 2017-2018 to examine surface currents and upwelling along and offshore the Vietnamese coast, using the concept of ocean intrinsic variability (OIV). A total of ten ensemble simulations have been conducted. In these ensembles, small-scale state of the initial field was perturbed in order to obtain a set of model ensembles without disturbing the whole forcing data (see Herrmann et al. 2023 for details). Subsequently, the SSC data obtained from these ensemble runs were used to calculate the ocean intrinsic variability index (OIV) and the monthly mean standard deviation of ensemble vorticity ($\bar{\sigma}_v$) from SSC field in April-May,





2017-2018. The OIV index, explained in Herrmann et al. 2023, is the ratio between the ensemble standard deviation and the ensemble mean of the temporal average over a given period of time of surface currents. It quantitatively assesses the contribution of OIV to the variability of the SSC over the whole period of analysis, in particular, April-May, 2017-2018. High values of $\bar{\sigma}_v$ were found near the coasts of the HFR domain, revealing large fluctuations within the ensembles induced by OIV (Fig. 12a-d). In terms of OIV index, high OIV were located parallel to the coasts, in particular approximately 15 km offshore along 12.9-13.1° N, associated with the northeast jet (Fig. 12e). This result emphasizes that the SSC variability is significantly influenced by OIV regardless external forcing. Similar conclusions and relevant discussion have been raised in previous studies for this region under different names as the North Coastal Upwelling (Ngo and Hsin, 2021; To Duy et al., 2022), South Vietnam Upwelling (SVU) (Da, 2018), Vietnamese coasts (Metzger, 2003).

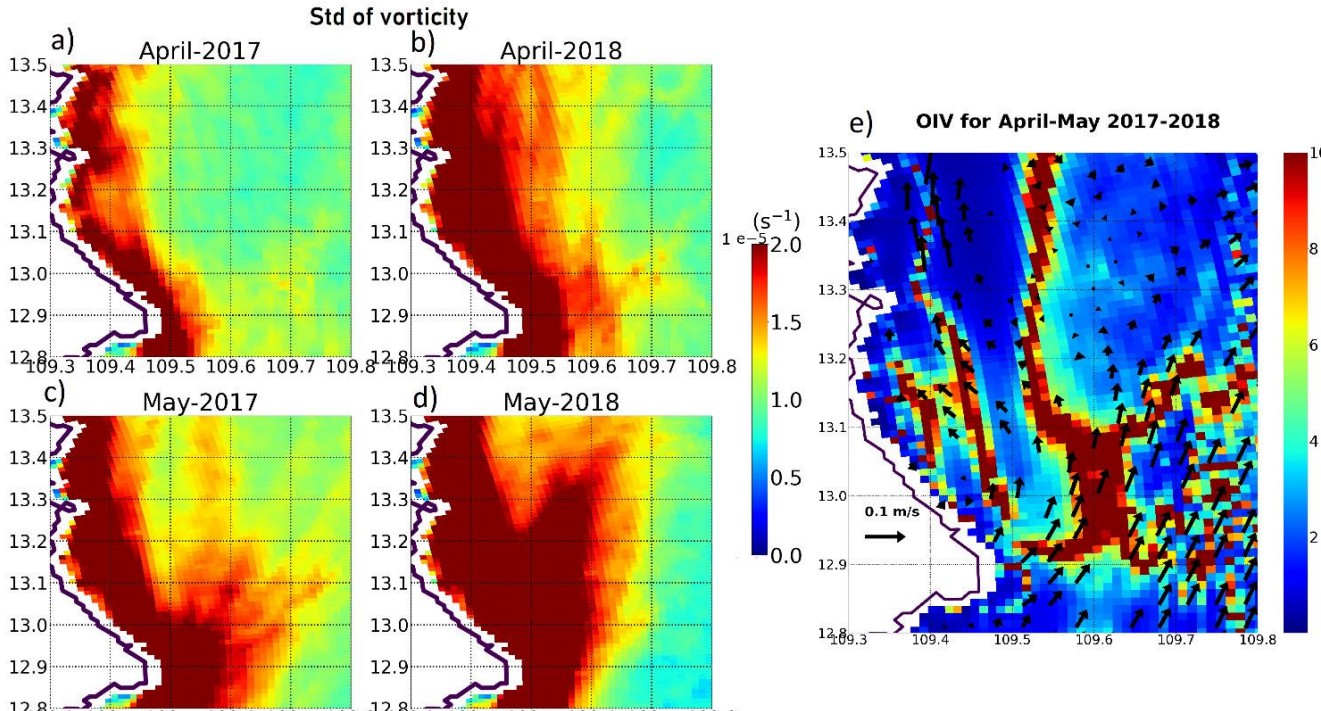

**Figure 12 (a-d) Monthly mean standard deviation of SSC vorticity over ensembles; e) OIV index superimposed with multi-annual monthly average SSC vectors for April-May, 2017-2018**

The study results above highlight that, winds play an important role in the variability of coastal surface currents in the VMSC at small temporal and spatial scales. Different wind forcing conditions using in different model simulations, i.e., VMSC_ref, VMSC_EkW, and VMSC_EnPS, resulted in significantly different SSC fields at an 1km resolution during a short time of analysis (May 5-15, 2019). As illustrated in Fig. 9e-f, opposite directions of SSC vectors between HFR and VMSC_ref were observed, which indicates that no northeast current occurred in the HFR domain during the time of analysis, whereas, this



strong northeast jet presented clearly in VMSC_ref simulation. Previous studies also pointed out that Ekman transport associated with winds can modify the position of the northeast jet and upwelling at the latitudes 12-14º N (Metzger, 2003; Shaw and Chao, 1994). Besides, De Gaetano et al. (2010) has confirmed the great impacts of different resolutions of wind on the inshore features of surface circulations, especially small-scale processes i.e. sub-mesoscale eddy formation. Thus, the limitation of grid resolution of the global atmospheric model can cause the incapability in capturing the local effects on wind

field i.e. orographic effects induced by complex terrains, and the coastal wind variability during the Monsoon transition period. In our case, two mountain ranges and a cape in the south of the domain (Fig. 1c) form a 'corridor' for winds to blow seaward in cross direction. The increase of surface heat at a valley between two mountains causes high convection and convergence zone at the coasts (Bei et al., 2018). Furthermore, when the activity of larger-scale winds is weak, this phenomenon becomes dominant. Counting the high wind variation during transition period, the behavior of coastal winds become more complex and

cannot be resolved at grid-scale of global models. Previously, a warning has been given in Metzger (2003) when using the ECMWF wind product in coastal model as spurious wind stress curls along the coastal boundaries, especially in regions with a complex orography, could reproduce misleading small-scale circulation along the coasts. Another point to be discussed is a poor availability of independent observed wind data for the coastal regions in Vietnam. The ASCAT wind data used for validation has a very coarse resolution (0.25 degree ~ 25 km) which is ambiguous for representing the wind field close to the

coastlines. The comparison between model runs and ASCAT data in Fig. 10 already shows that the behaviors of winds amongst different wind time-series are more comparable at far field, where the land morphology and orography no longer have impacts on coastal dynamics.

Both EnPS and EkW methods have constraints in producing better wind field for the whole analysis period. At mid field and far field, the EnPS method achieved better wind field compared to ECMWF wind data however large discrepancies still

occurred during the period when the ECMWF showed good agreements with ASCAT winds. This could be explained by the fact that nearshore circulations (the first ~30 km offshore) are not only governed by winds but other processes, as explained above. Besides, the optimization methods performed better during southern and southwestern winds, and is sensitive to the change in wind direction and when wind speed is weak. This constraint could be due to the insufficient number of ensembles used in EnPS method. In this research, only a cluster of 50 wind forcing ensemble members were used while Barth et al. (2011)

used 100 wind ensemble members for correcting the model surface currents to obtain good statistics for calculating covariances. In case of EkW method, a constant drag coefficient was used instead of wind-dependent one. The sensitivity to the drag coefficient variation has not yet been tested.

**6 Conclusion and future work**

The coastal dynamics of the VMSC, especially within the HFR measurement domain, during the transition period of summer

Monsoon is influenced by various factors, amongst others, winds, coastal topography, and larger-scale background circulation variability. Compared to SSC obtained from HFR measurements, SYMPHONIE model is capable to characterize the coastal





circulation variability as it could capture relatively well the spatial patterns of SSC. However, a large discrepancy in velocity time series between model and HFR are found in April associated with the eastern and southeastern winds. This is a limitation of model in simulating coastal circulation under complex variation of wind.

The EnPS and EkW methods show improvements in v component of SSC as RMSE and MAE reduced remarkably (36-40 %), whereas, there is no significant improvements in u component. The bottleneck of this study is there are no nearby offshore wind stations or wind measurements during the HFR measurement period for validation. Wind data records at two nearest meteorological stations are not representative as the stations are located inland far from the shore. The ASCAT wind data used for validation have a very coarse resolution thus cannot describe the wind behavior due to orographic effects close to the coasts.

Therefore, a comprehensive measurement campaign including wind measurements should be considered in the next step of our future work on the region.

A complex variability of the coastal dynamics in the VMSC includes a substantial contribution from non-linear interaction, i.e., oceanic intrinsic variability (OIV). Although the overall discrepancies were reduced by ~40 %, a good agreement between the SSC time-series obtained from HFR measurements and from wind-optimized model runs is still not yet established.

Especially during calm wind conditions, strong southeastward current velocities still presented all over the HFR domain. This phenomenon could be governed by different processes, for instance, the background circulation at larger scales from outside of the VMSC domain. Thus, wind optimization alone is not capable to deliver a sufficient improvement in simulation results of surface currents. For this reason, OBC optimization should be tested in the next step of our study with an expectation to obtain a better model simulation of the coastal dynamical processes in the VMSC region.

Last but not least, the optimisation results revealed that the surface circulation is influenced not only by winds but also by other factors such as intrinsic ocean variability which is not entirely controlled by boundary conditions. This indicates the potential usefulness of large velocity datasets, amongst others, an adequate monitoring network with good coverage and temporal resolution, i.e., HFR systems, and other data fusion methods to effectively improve modelling results in such highly-dynamic regions.

## 7 Appendix

Figure A1 shows accuracy values which were extracted from standard WERA Data Viewer provided by the HFR manufacturer. Higher uncertainties of both u and v components of SSC from HFR measurements are found in the offshore range starting from a distance of ~40 km. The high values (high noise level) are associated with the occurrence of 50 Hz noise band and its harmonics caused by electrical power supply in the surrounding areas of WERA locations. Spurious radial currents are the

results of the smeared-out signal of noise with the first-order peaks (which give the information of radial currents) in the Doppler spectra. Although most of spikes in original HFR data have been removed, the noise still contaminated some parts of the domain, in particular at range more than 40 km, where 50 Hz-harmonic noise bands are present.



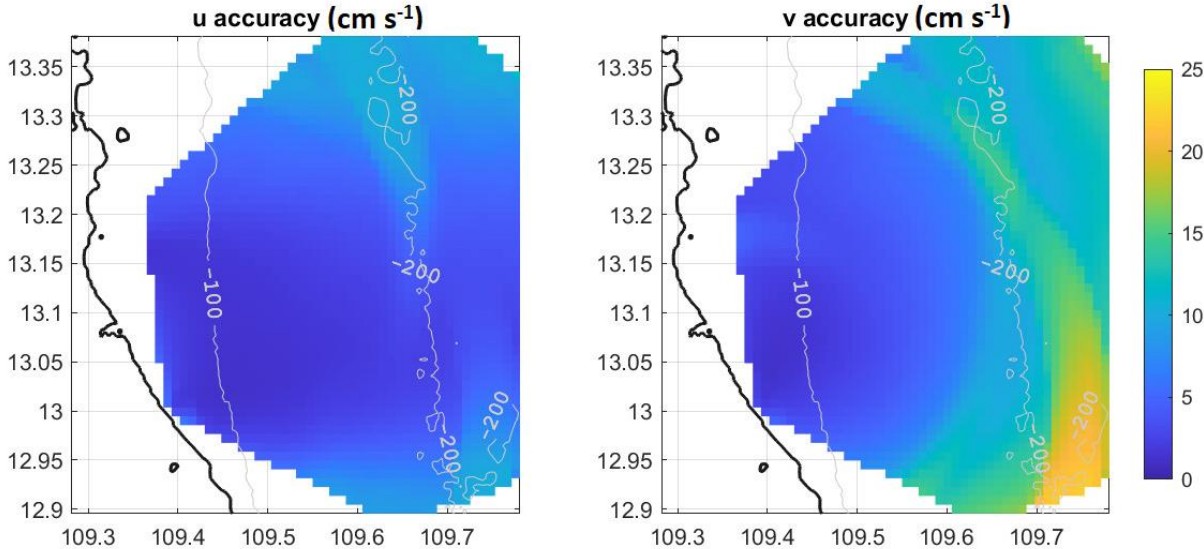

**Figure A1:** Mean accuracy map of u and v components of sea surface currents from HFR measurements for the entire period of measurements.

## Code and data availability

The SYMPHONIE model can be downloaded free-of-charge from https://sirocco.obs-mip.fr/ocean-models/s-model/download/ and is presented in Marsaleix et al. (2008). Atmospheric forcing data and ocean data used as initial ocean conditions and lateral ocean boundary conditions are available on https://10.5281/zenodo.13253326.

Measurement data (HFR and AWAC) should be requested from CEFD-HUS/VNU and cannot be provided by the author.

The code of optimization methods can be provided upon request.

## Author contribution

TDT and MH built the model configuration and ran the simulation from 2009-2018. MH provided model forcing data and ensemble outputs for OIV analysis. AS, SO and THT worked on the optimisation and ocean analysis methods, and designed the experiments. KCN provided measurement data from HFR and was responsible for the project fund. THT prepared the paper and collected all contributions from other authors.

## Competing interests

The authors declare that they have no conflict of interest.



**Acknowledgements.** This research was funded by the joint Doctoral's Program between the University of the Littoral Opal Coast (ULCO) and University of Science and Technology of Hanoi (USTH), and partially funded by the Vingroup Innovation Foundation (VINIF) under project code VINIF.2023.DA151. It is also supported by the LOTUS international joint laboratory program (lotus.usth.edu.vn). The HFR data were collected from the measurement campaign in Phu Yen, 2019 and provided

by the Center for Environmental Fluid Dynamics, University of Science, Vietnam National University (CEFD-HUS/VNU).

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
