# Peer review of "Surface circulation characterization along the middle-south coastal region of Vietnam from high-frequency radar and numerical modelling"

_EGUsphere, 2024_

## Author Response (AR1)

| ID | Reviewer's comments | Authors' responses |
| --- | --- | --- |
| RC1#1 | "the spatial and temporal evolution of the surface circulation, but some discrepancies were found between model and HFR data on some days, coinciding with the evolution of the wind. Two methods were used to optimise the wind forcing, namely the Ensemble Perturbation Smoother (EnPS) and the wind correction method using wind-driven surface currents (EkW)" this is assuming that the model is not following wind forcing, or wind forcing is not accurate enough. Do you have independent wind data from meteorologic stations nearby to further confirm this? | Yes, there are two meteorological stations nearby. However, they are not representative for the wind in the open sea since they are located far inland. The additional text to explain this was elaborated in lines 405-409 in Section Discussion. |
| RC1#2 | "The optimisation results revealed that the surface circulation is not only driven by winds but also by other factors such as intrinsic ocean variability which is not entirely controlled by boundary condition". what are these processes and factors? | The main factor was a nonlinear behavior of the ocean that has been explained by a concept of ocean intrinsic variability, explained in lines 366-375 in the Section "Discussion"  Another factor was the interaction of a powerful flow with headlands. This was added in the "Conclusion and future work" Section, in lines 446-448 |
| RC1#3 | L33, "and fully" "and is fully" | The text was modified |
| RC1#4 | L40:"coastal dynamics along the VMSC is strongly influenced by ocean intrinsic variability" such as? | Such as: producing variations of upwelling expansion, location of sub-mesoscale eddies and current jets, and the intensity and size of eddies The text in lines 40-41 was modified to meet the Reviewer's recommendation. |
| RC1#5 | L90: remove "performed" | The text was modified |
| RC1#6 | L104-108: explain more in details how data analysis is performed: what depth the current meter data refer to, what grid point is used the the radar data, what data QC is used for the current meter,…and so forth. Data quality from the current meter seems to be failry poo if compared to HFR currents. what is the casue for that difference? is it maybe using the closest bin to surface which may be contaminated? Provide units for | Explanation to your questions:  1) The fairly poor quality of AWAC measurements was due to the impact of waves since the data was taken from the surface layer depth (1.5m bin size). 2) The discrepancy between AWAC and HFR velocity time-series can be attributed to two primary factors: the difference in measurement depths and the wave-induced effects on AWAC measurements in the surface layer depth. 3) Regarding the baseline, the surface current used in all analyses have been reconstructed |

| ID | Reviewer's comments | Authors' responses |
|---|---|---|
| | MAE and RMSE. The current meter is at the oundary of the V2 station and fairly close to the baseline so errors and differences may be explained by a combination of factors | by using 2DVar method, a non-local interpolation technique, providing good-quality vector maps, also in areas with high GDOP (geometric dilution of precision).

The text in lines 105-115 was modified to meet the Reviewer's recommendation |
| RC1#7 | L205: "VMSC_ref's time-series associated with wind time-series, but not for HFR measurements." the inertial peaks are also found on the model simulations. Interestingly, model and HFR spectra share same structure in the low freq band as the wind suggesting that this band is relatively well mateched however this is not found in the high-frequency tails. I remember seeing this elsewhere where models and HFR data were compared and that was explained by the poor wind energy in these frequency band, the model restart or the lack of stratification in the model. how does this fit within the context of the region of interest here? | Regarding the high-frequency tails, two reasons have been explained in lines 224-229. The explanations for the coherence between the model and observation within sub-tidal bands, as well as the impact of low-resolution forcing data on the model's ability to reconstruct higher-frequency variabilities of surface circulation, have been elaborated in the revised manuscript, which includes a text and two new references (lines 229-234). |
| RC2#1 | While the use of HFR data to correct winds seems to provide a very promising approach for models, HFR measures the total velocity including the stokes drift from waves. I think that in this case, Eq 7 is not valid.

Can the authors explain this point? | We quantified the contribution of sea states (waves) to the surface current obtained from HF radar using equations A4 and A5 in the paper of *"A. Sentchev, P. Forget, Y. Barbin, M. Yaremchuk, Surface circulation in the Iroise Sea (W. Brittany) from high resolution HF radar mapping, Journal of Marine Systems 109-110 (2013) S153–S168"*.

The space-averaged wind speed from the ECMWF did not exceed 6-7 ms$^{-1}$, and the significant wave height ($H_s$, space-averaged) did not exceed 0.5 m. With this information, we proceeded to quantify the velocity of wave-induced currents, whereby we determined that the contribution of Stokes-drift to the total surface currents measured by HF radar was estimated at 0.02 m/s, representing approximately 4-5% of the total surface current velocity.

This illustrates that the Stokes' impact on the present velocity estimation from the EkW |

| ID | Reviewer's comments | Authors' responses |
|---|---|---|
| | | method can be neglected. We're assuming that Eq. 7 applies. |
| RC2#2 | The simulation experiments for the wind reduction (from the 5th - 19th May) is too short to infer some conclusion about the general improvement of the methodology. Are they just resulting from the specific wind and mesoscale conditions present during the transition phase? | We have applied the methods to different periods (both April and May) but only selected the specific time period from May 5 to May 14 (10 days). We agree with the Reviewer that the selected period was short. However, during this period, the current velocity maps from HFR demonstrated large variability of circulation patterns (please see Fig. 6a,b,c), which were not consistent with the evolution of the ECMWF wind. Additionally, a significant discrepancy was observed between the model and observations in V component of the surface currents during this period (Fig. 7e,f). With all those evidences, our hypothesis was that the wind forcing was the main cause of errors in model simulations. During this period, there was a significant shift in the wind direction, which could potentially explain the identified errors. |
| RC2#3 | I don't see the necessity of including Section 3.4 in this manuscript. | We could not meet the Reviewer's recommendation because we do not have the Section 3.4 in our manuscript. |
| RC2#4 | Page 19. line 351. Remove "nonlinear chaos" | The text in lines 361-366 in the revised version of the manuscript was modified toward the comment of the Reviewer. |

---

## Author Response (AR2)

| ID | Editor's comment | Authors' response |
|---|---|---|
| EC#1 | I think that you should include in the revised manuscript at least a summary of your response to referee comments RC2#1 and RC2#2. Readers of a final paper may have the same questions and it is much better to have the answers in the paper than for the reader to have to hunt in the comments for an answer. | Thank you for your precious suggestion. The summary of the RC2#1 and one reference have been elaborated in the revised manuscript in lines 178-182 The summary of the RC2#2 has been elaborated in the revised manuscript in lines 197-199 |
| EC#2 | In some places you mention "non-linear interaction" but I am not sure that is the best description. For example, lines 437-438 "non-linear interaction, i.e., oceanic intrinsic variability (OIV)" I think "oceanic intrinsic variability (OIV)" would suffice. If you do refer to non-linear interaction, you should name at least two processes that are interacting. | The lines have been removed from the revised version of the manuscript due to repetition in other lines (refer to EC#6) |
| EC#3 | Lines 224-225. This sentence is not clear to me. I understand the sentence in lines 231-232 but in line 225 "this behaviour in PSD spectrum" is too vague: what behaviour in the spectrum of which (estimate of) velocity? One would expect coarse resolution to result in very little high-frequency power, i.e. a steep slope at high-frequency bands. | In the revised manuscript, the authors have provided additional clarification and one new reference regarding the sub-tidal range observed in the PSD spectrum of wind speed, as detailed on lines 217-220. Furthermore, an explanation and two new references have been provided regarding the discrepancy between the PSD spectra of wind speed and current velocity, which |

| | | can be found on lines 237-242. |
|---|---|---|
| EC#4 | Line 253. "we investigate the multiannual variability of SSC" but figure 5 is a ten-year average and shows only spatial variability, nor is there any description in the text. | The text has been modified toward Editor's comment (lines 268-269 in the revised manuscript) |
| EC#5 | Line 399. "in cross direction" is unclear until you state "cross-coastal" or "cross-shelf" or something similar. | The text has been modified toward Editor's comment (lines 412-413 in the revised manuscript) |
| EC#6 | Lines 437-438 and 445-446. Some repetition here. | The lines 437-438 in older version of the manuscript have been removed from the revised version. |

---

## Author Response (AR3)

| ID | Editor's comment | Authors' response |
|---|---|---|
| EC#1 | In a few places you refer to "southern" winds and similar. It is better to use "southerly" if the wind is from the south (meteorological convention) and "southward" if the current is towards the south (oceanographic convention). "southern" is better used if the location of the measurement is in the south. | All texts regarding directions of winds and currents have been corrected toward meteorological and oceanographic conventions. |
| EC#2 | Line 122. "using Leap Frog scheme combined Laplacian filter" is not very clear. I think at least "using a Leap Frog scheme combined with a Laplacian filter" but maybe more is missing. If this scheme is not in Marsaleix et al., 2008, then I think another reference is needed. | One reference (Marsaleix et al., 2012) have been added to the revised manuscript and text in lines 123-124 was modified in the revised manuscript for a clearer explanation. |
| EC#3 | Lines 122-123. "K-epsilon turbulence closure scheme, and pressure gradient Jacobian scheme to describe the physics within the model." Are these schemes in Marsaleix et al., 2008 or is another reference needed? | The schemes were already explained in Marsaleix et al., 2008. |
| EC#4 | Lines 205-210. I think there needs to be some more definition of VMSC_ref. Perhaps "free" is misleading. Please be explicit. [At least I suppose there must be some form of boundary condition for the model to run. Is there any wind or tidal forcing? Lines 223, 220-221 and section 3.2 suggest "yes". But then why do the EASTSEA and VMSC_ref series for U and V differ in Figure 7? This is discussed in the Discussion section 5 where the VMSC_ref open boundary conditions and OIV are suggested as causes. That rather implies that the wind and tidal forcing of EASTSEA and VMSC_ref are the same; | We thank the Editor for a valuable comment. The "free run" text has been removed from the revised manuscript to avoid misunderstanding. Indeed, the forcing (wind, tide, river) were used in all simulations of VMSC_ref as well as in EASTSEA run. It has been explained in lines 129-134 of the revised manuscripts. So VMSC_ref run is not free run. We are sorry for the improper use of |

| | | |
|---|---|---|
| | VMSC_ref is not really a "free" run.] | the term. |
| EC#5 | Line 356. "latter days" is unclear. Please be explicit with a date that relates to figure 11. | The text in lines 357-359 has been modified toward the Editor's comment. Labels (a-f) have been added for all subplots in Figure 11. |
| EC#6 | Line 435. Better ". . data; however, large discrepancies . ."? | The text has been modified toward the Editor's comment |